# Iron Dysregulation and Inflammagens Related to Oral and Gut Health Are Central to the Development of Parkinson’s Disease

**DOI:** 10.3390/biom11010030

**Published:** 2020-12-29

**Authors:** Marthinus Janse van Vuuren, Theodore Albertus Nell, Jonathan Ambrose Carr, Douglas B. Kell, Etheresia Pretorius

**Affiliations:** 1Department of Physiological Sciences, Faculty of Science, Stellenbosch University, Private Bag X1 Matieland, Stellenbosch 7602, South Africa; mjvanvuuren@sun.ac.za (M.J.v.V.); tnell@sun.ac.za (T.A.N.); 2Division of Neurology, Department of Medicine, Faculty of Medicine and Health Sciences, Stellenbosch University, Private Bag X1 Matieland, Stellenbosch 7602, South Africa; 3Department of Biochemistry and Systems Biology, Institute of Systems, Molecular and Integrative Biology, Faculty of Health and Life Sciences, University of Liverpool, Crown Street, Liverpool L69 7ZB, UK; 4The Novo Nordisk Foundation Centre for Biosustainability, Technical University of Denmark, Building 220, Chemitorvet 200, 2800 Kongens Lyngby, Denmark

**Keywords:** Parkinson’s disease, bacteria, lipopolysaccharides, iron, gingipains, amyloid and α-synuclein

## Abstract

Neuronal lesions in Parkinson’s disease (PD) are commonly associated with α-synuclein (α-Syn)-induced cell damage that are present both in the central and peripheral nervous systems of patients, with the enteric nervous system also being especially vulnerable. Here, we bring together evidence that the development and presence of PD depends on specific sets of interlinking factors that include neuroinflammation, systemic inflammation, α-Syn-induced cell damage, vascular dysfunction, iron dysregulation, and gut and periodontal dysbiosis. We argue that there is significant evidence that bacterial inflammagens fuel this systemic inflammation, and might be central to the development of PD. We also discuss the processes whereby bacterial inflammagens may be involved in causing nucleation of proteins, including of α-Syn. Lastly, we review evidence that iron chelation, pre-and probiotics, as well as antibiotics and faecal transplant treatment might be valuable treatments in PD. A most important consideration, however, is that these therapeutic options need to be validated and tested in randomized controlled clinical trials. However, targeting underlying mechanisms of PD, including gut dysbiosis and iron toxicity, have potentially opened up possibilities of a wide variety of novel treatments, which may relieve the characteristic motor and nonmotor deficits of PD, and may even slow the progression and/or accompanying gut-related conditions of the disease.

## 1. Introduction

The global prevalence of Parkinson’s disease (PD) during 2016 reached 6.1 million [1]. In the United States of America, Canada, and Europe the prevalence is projected to increase by approximately 92% by 2050 [2], involving an increased burden on global healthcare [3].

α-Synuclein (α-Syn) is the principal component of Lewy bodies (LBs), which are the pathological hallmark of PD and other related conditions [4]. This group of illnesses, termed synucleinopathies, includes multisystem atrophy, dementia with Lewy bodies, and pure autonomic failure. α-Syn-stained inclusion bodies and fragments of neurons are detectable in many otherwise healthy individuals at the time of postmortem examination and detailed histological examination has led to conclusion that the initial lesions of PD are thought to occur in the medulla, in the region of the dorsal motor nucleus of the vagal nerve, and also in the olfactory bulb [5,6,7]. In PD patients, LBs are also observed in nondopaminergic neurons outside of the basal ganglia, in areas such as the glossopharyngeal–vagal complex, coeruleus–subcoeruleus complex, caudal raphe nuclei, gigantocellular reticular nucleus, and olfactory pathways [5]. The knowledge that LBs consist mainly of misfolded amyloid forms of α-Syn has, therefore, shaped investigations in the field of PD research to adopt a “synuclein-centric and neuro-centric” approach, predominantly focusing on the central nervous system (CNS). However, recent studies have challenged this neurocentric approach [8]. Neuronal lesions in PD, commonly associated with α-Syn-induced cell damage, are, therefore, present both in the central- and peripheral nervous systems (CNS or PNS) of PD patients [9], with the enteric nervous system also being especially vulnerable [10].

Of particular interest in PD is the prion hypothesis in which it is postulated that fragments of the protein α-Syn may lead to the formation of progressive accumulation of amyloidogenic protein material [11,12]. In accordance with this theory, PD is characterized by a caudo-rostral progression of deposition of α-Syn, associated with neuronal loss, and with positive staining for α-Syn of LBs and Lewy neurites [6]. Involvement of both the olfactory bulb and the medulla is explained by a proposed dual-hit mechanism, involving anterograde progression of pathology from the olfactory system into the temporal lobe, and retrograde progression to the brainstem from the gut following ingestion of a neurotropic pathogen [13,14]. α-Syn staining material is present in patients with PD in a number of tissues outside the brain and spinal cord and is present before the diagnosis is made [15,16]. It is also noteworthy that nonmotor features such as rapid eye movement (REM) sleep behaviour disorder [17], constipation [18], loss of the ability to smell, and sleep disorders may precede the appearance of tremor and other classical features of PD by as much as two decades or more. Gastrointestinal symptoms are common and can precede motor symptoms and clinical PD diagnosis [19].

Although PD research and clinical perspectives have focused on genetics and efforts to remove or prevent the distinctive α-Syn-containing cytoplasmic inclusions (LBs), little progress has been made to prevent progression of disease or to cure PD. A number of PD genes have been identified as playing a role in PD aetiology [20]. Even collectively, mutations in these six genes explain only a limited number (3–5%) of sporadic disease occurrences [21]. (For an overview of PD genes and dopaminergic neurons, see [20].) Over the past years, researchers have published significant numbers of papers, which suggests that the motor effects and the progressive degeneration and loss of preferential dopaminergic neurons in the substantia nigra seen in PD patients, might be related to factors other than a predominantly genetic cause. Recently, there has been increasing focus on investigating the link between increased iron and copper levels (both systemically and in the brain) [22], the role of the microbiome and how dysregulated circulating inflammatory biomarkers, including those originating from bacteria, might all interplay in PD pathology [19].

In this paper, we aim to bring together evidence that the development and presence of PD depends on specific sets of interlinking factors. Together with neuro-inflammation, patients with PD also have systemic inflammation, with many cellular signals pointing to major vascular dysfunction. We discuss published evidence that PD is both associated with and driven by dysregulated circulating inflammagens (an irritant that elicits both oedema and a cellular response of inflammation) and cytokines. Furthermore, there is significant evidence that bacterial inflammagens fuel this systemic inflammation and might be central in the development of PD [3,23,24,25,26,27,28]. There are a number of papers suggesting altered immune activation in PD [29,30,31,32] and also, papers demonstrating altered immune activation in PD populations [33,34,35]. There are also experimental works indicating that inflammatory action can induce PD-related phenotypes [36,37,38,39]. The immune response in PD, which is proposed to occur early, involve peripheral and brain immune cells, eventually evolve as neuronal dysfunction progresses, and is likely to influence disease progression [40]. Central to these immune changes in PD is the interplay between the microbiome–gut–brain axis [25,41,42,43]. 

Since these changes are taken to be deterministic [44], we furthermore point out that, together with genetic predisposition and epigenetic changes, the only way to address this extremely significant healthcare dilemma is to find the origin of these inflammagens and the reason for immune activation in PD. We argue that this origin is microbial. Central to its cause might be microbiome dysregulation, translocation, and comorbidities including periodontitis and gingivitis. For an overview, see Figure 1. 

## 2. Parkinson’s Disease, Iron, Oxidative Stress, and Chronic Systemic Inflammation

Chronic inflammation, the circulation of pro-inflammatory molecules, and the dysregulation of innate immune responses can contribute significantly to both the onset and progression of neurodegeneration in PD [32,45,46,47,48,49,50,51,52]. The spectrum of iron dysregulation in PD includes iron storage, uptake, and release [53], and importantly, the effects may be seen both in the brain and also the peripheral tissues. Iron dysregulation in PD, occurs both in the brain and in circulation (Figure 1) (for a review, see [54]). It was previously shown that increased serum ferritin levels are present in PD, where it may directly cause eryptosis of erythrocytes [54]. In circulation, iron dysregulation is directly implicated as a major cause of oxidative stress [44,55]. Furthermore, circulating iron and serum ferritin dysregulation (due to the release of poorly liganded iron) causes pathological changes in both erythrocyte and fibrin morphology [56,57,58,59]. In 2012, it was also reported that serum ferritin levels were significantly increased in male and female PD patients and were correlated with PD severity stages and duration in men and women [60].

Iron dysfunction in the substantia nigra is considered one of the fundamental reasons for dopaminergic neurons dysfunction and death [20,61,62,63,64]. Recently, ferroptosis (a kind of regulated cell death that is characterized by highly iron-dependent lipid peroxidation) has been associated with neurodegenerative conditions [65,66] and, particularly, PD [67]. Ferroptosis happens due to the depletion of plasma membrane unsaturated fatty acids and accumulation of iron-induced lipid ROS [68]. The overaccumulation of lipid ROS leads to an oxidative stress response in cells that causes lethal damage to proteins, nucleic acids, and lipids and eventually to cell death [66,69]. For a comprehensive review on ferroptosis in PD, see [70]. An important iron compound in dopamine and norepinephrine neurons is the neuromelanin–iron complex [53,71]. Neuromelanin is a complex polymer pigment found primarily in the dopaminergic neurons of human substantia nigra, and it is stored in granules including a protein matrix and lipid droplets [72]. Interestingly, neuromelanin is a strong iron chelator [53]. In PD, there is a loss of neuromelanin in the substantia nigra, and this could lead to enhanced calcium messaging, followed by formation of reactive oxygen species (ROS), and eventually neuronal apoptosis [73].

Serum ferritin is also upregulated in the circulation of PD individuals [54]. It has been well documented that the absence of free iron restricts the growth of pathogens. Here, we point out the corollary—that the presence of free iron allows their proliferation (and hence translocation). In particular, the cytotoxicity caused by invading microbes causes the release of iron that allows their (limited) proliferation and the release of more inflammagenic bacterial products. 

Figure 2 shows the important role of iron, mitochondria, and lysosomes in substantia nigra neurons (adapted from Funke and coworkers (2013)) [20]. Physiological levels of iron and the presence of oxidative stress are present not only in the brains of PD patients but also in their circulation. In PD patients, excessive systemic inflammation also occurs [24,45], simultaneously exacerbating and perpetuating neuro-inflammation and may also triggering of the inflammatory events ongoing in the brain. There is an increasing recognition of the involvement of Toll-like receptors (TLRs) in neuronal degeneration as cells of the nervous system express TLRs and these TLRs are activated by α-synuclein. The activation of these TLRs induces an inflammatory response that precedes neuronal loss [25]. In addition, interleukin-1 (IL-1), can be synthesized and released by activated microglia, and could possibly cause dopaminergic neurodegeneration leading to PD [74]. Activated microglial cells have also been shown to be involved in the secretion of TNF-α thereby contributing towards the progress of PD [75]. In addition, α-Syn also induces TNF-α [76]. 

Pathological levels of iron and oxidative stress are also central to the persistence of neuroinflammation. A cycle of decreased levels of endogenous antioxidants, increased ROS, augmented dopamine oxidation, and high iron levels have been found in brains from PD patients [77]. Examples are decreased presence of coenzyme-Q10 (CoQ10), uric acid, and vitamin E [78]. CoQ10 scavenges free radicals, with main function of protection of mitochondrial and lipid membranes [79]. Uric acid, which acts as an antioxidant, is also lower in PD [79]. Vitamin E is known to prevent lipids from oxidative stress, and plasma levels of vitamin E are reduced in PD patients [80]. Dopamine metabolism, high levels of iron and calcium in the substantia nigra, mitochondrial dysfunction and neuroinflammation directly contribute to the increased oxidative stress and dopaminergic neuronal loss in the brains of PD patients [78,81]. In a 2016 study, serum levels including that of iron, ferritin, transferrin, superoxide dismutase catalase, nitrosative stress marker, thiobarbituric acid reactive substances, and other similar oxidative stress markers were analysed in 40 PD patients and 46 controls [82]. The authors concluded that ROS/RNS production and neuroinflammation may dysregulate iron homeostasis and that oxidative stress may be a key driver in the pathophysiology of PD. See Table 1 for selected references regarding systemic inflammatory mediators and numerous papers reporting on a dysfunction of iron metabolism in PD. 

## 3. An Integrated Physiological Systems Disease

The question that now arises is if there is any substantial evidence that pathological iron levels, and increased presence of inflammatory biomarkers (both in circulation and the brain), could originate from somewhere else than the PD brain. The quest for the origin and the trigger of PD pathology, has led researchers to search for systemic hallmarks that characterize PD patients [45,88,89,90,91,92,93]. Data suggest that most PD patients may suffer from gut dysbiosis and other conditions that allow bacterial translocation. The origin of dysregulated circulating inflammatory biomarkers could therefore involve bacteria, and particularly their inflammagens, entering the body via gut dysbiosis and translocation (when microbes appear in places other than their normal location) [19]. These inflammagens might in fact contribute significantly to the increased iron levels and circulating cytokines in PD [39,61,64,76,116]. 

### A Continuum of Pathological Events or the Distinct Events Hypothesis?

The aetiology of PD is a complex process and the evidence in persons with PD (with or without REM behavioural disorder) indicates that degeneration may start either in the CNS or in the peripheral nervous system (PNS) [54], affecting numerous fundamental cellular processes [117]. A continuum of pathological events or the “chicken or egg” hypothesis is, therefore, not a straightforward assumption. A recent 2019 paper discusses the hypothesis of brain-first or gut-first in the development of PD, where the authors argue that it is hypothesized that PD can be divided into various subtypes, as either PNS-first and a CNS-first subtype [118]. Central to the extent of involvement of bacteria, and the presence of gut dysbiosis and translocation in PD, is the question of autonomic dysfunction, i.e., orthostatic hypotension, constipation, bladder disturbances, and sexual dysfunction [119]. These symptoms precede neuromotor symptoms and are strongly associated with impaired activities of daily life and dementia during later stages of the disease [120,121]. Dysautonomic symptoms are frequently found in the various α-synucleinopathies, including PD [122]. Autonomic dysfunction affects the enteric nervous system, resulting in constipation [26]. An important concept to consider, is whether dysbiosis and translocation are causes of autonomic dysfunction and α-synucleinopathies or whether neuro-inflammation causes a spill-over peripheral autonomic dysfunction. Evidence that might provide an answer to this critical question may be found in the timing of the various pathologies. Already in 2001, Abbot and coworkers suggested that further studies are needed to determine whether constipation is part of early PD processes or is a marker of susceptibility or environmental factors that may cause PD [18]. In PD patients, α-Syn inclusions have been detected in colon biopsies up to 8 years before the onset of motor symptoms of PD, and it has been argued that the presence of extracellular α-Syn is associated with acute and chronic inflammatory conditions of the intestine [123,124]. It may be worth noting here that individuals without PD may also have intestinal α-Syn inclusions, which can also be detected [125]. The specificity and sensitivity of colonic biopsies for the detection of pathological α-Syn inclusions is, therefore, conflicting across various research. A recent prospective cohort study also showed that patients presenting with pure autonomic failure are at high risk of phenoconverting to a manifest neuro-synucleinopathy [126]. Sampson et al., (2016) demonstrated that faecal transplantation of gut microbiota from PD patients enhanced α-Syn-mediated motor dysfunction in a mouse model of PD [127]. The authors further found that germ-free animals displayed significantly fewer α-Syn inclusions in the CNS, indicating that gut microbiota are required for α-Syn aggregation and formation of the hallmark inclusions seen in PD. 

In their hypothesis, Borghammer and Van Den Berge (2019) [118] suggest that PD associated with the PNS-first hypothesis is tightly associated with REM sleep behaviour disorder during the prodromal phase and is characterized by marked autonomic damage before involvement of the dopaminergic system. In contrast, the CNS-first phenotype is most often REM sleep behaviour disorder negative during the prodromal phase and characterized by nigrostriatal dopaminergic dysfunction prior to involvement of the autonomic PNS [118]. The current available evidence not only points to intestinal involvement but also it should be highlighted that there is an absence of longitudinal research studies. Such studies need to be conducted to provide appropriate data with strong evidence to support this hypothesis. According to our understanding of the sequence of events, evidence points to an intestinal involvement, followed by systemic inflammation and then the occurrence of neurological PD symptoms (see Figure 3). It can furthermore also be presumed that many diseases, including PD, can be caused by a positive feedback, and that this might lead to overwhelming of homeostatic mechanisms. In this case, the positive feedback mechanism (namely, A causes B then B causes more A, etc.) is dysbiosis/translocation and amyloid formation and deposition. In Figure 3, we argue that this positive feedback mechanism is present in PD, and it is between α-Syn aggregation and autonomic dysfunction. 

## 4. Oral and Gut Dysbiosis and Parkinson’s Disease

There is significant evidence of gut dysbiosis in PD, as reviewed in the following paragraphs. The gastrointestinal tract can communicate with the central nervous system by several mechanisms, including hormones, cytokines, and microbial metabolites via circulation, as well as direct neural circuits via the autonomic nervous system (vagus nerve) [128]. Dysregulation of the bidirectional signalling system (the gut–brain axis, also known as the gut–microbiota–brain axis or the microbiome–gut–brain axis), is well-known in PD. Parkinson’s disease patients have a significantly higher incidence of comorbid gastrointestinal dysfunction, with between 60% and 80% of patients suffering from constipation [129] and intestinal inflammation [10]. Gastrointestinal dysfunction is, therefore, a very well-known accompaniment to PD [130,131] and also precedes the onset of motor symptoms by several years [132]. It has been shown that faecal and mucosa-associated gut bacteria of PD patients differ substantially from healthy individuals [103,132,133]. Nuzum et al., (2020) review literature that discussed gut microbiota differences between PD groups and controls, and suggested that were variations may potentially be the cause of PD pathophysiology [19]. However, the authors noted that differences in methodologies may be problematic [19]. Parkinson’s disease-associated constipation has also been found to correlate with α-Syn accumulation in the enteric nervous system, resulting in local inflammation, oxidative stress, and increased intestinal permeability [134,135]. A significant body of work has, therefore, implicated gut dysbiosis as a major contributory factor to the constipation observed in PD patients. 

α-Syn aggregation may originate in the gut and propagate via the vagus nerve to the brain. This hypothesis is supported by reports of α-Syn inclusions being observed in the enteric nervous system, including the vagal nerves [7], sometimes years before the onset of the first motor symptoms [136]. Injection of α-Syn fibrils into the gut tissue of healthy rodents also induced pathology in the vagus nerve and brainstem, indicating that misfolded α-Syn propagates from the gut to the brain via the vagus nerve [137]. It has also been shown that vagotomized subjects are at a lower risk of developing PD [138]. When gut dysbiosis is present in PD patients, gut-derived bacteria escape from the gastrointestinal tract into the blood. This process leads to shedding of endotoxins into the systemic circulation, which may constitute a trigger event in the development of PD and other neurodegenerative disorders [45,139]. Moreover, there is also evidence for bidirectional and trans-synaptic parasympathetic and sympathetic propagation of alpha-synuclein in animal models [140]. However, these models report varying results [141,142,143]. 

### 4.1. Bacterial Inflammagens

Gut dysbiosis is typically associated with an increase in Gram-negative bacteria such as *E. coli* and *H. pylori*, which are known to secrete a variety of pro-inflammatory molecules [135,144]. Bacterial inflammagens can act as both cytotoxins and neurotoxins, which disrupt the homeostatic functioning of cells in circulation and tissues [145,146]. Microbes shed their respective inflammagens (endotoxins) in response to different physiological and environmental cues, potentially resulting in a broad spectrum of deleterious effects [44,147]. These include both systemic inflammation and neuro-inflammation as well as impaired gut barrier function [44,148]. Bacterial inflammagens implicated in inflammation may include proteolytic enzymes such as carbonic anhydrases, peptidyl deiminases and gingipains, as well as bacterial appendages like curli fibres and fimbriae, LPS, or lipoteichoic acid (LTA) [102,149,150,151,152]. There is also evidence that curli can also accelerate synuclein pathology in rat and worm models [153] and in mice models [116]. Many bacteria can also assemble functional amyloid fibres on their cell surface and these amyloids contribute to biofilm formation where cells interact with a surface or with other cells [154]. Furthermore, these bacterial amyloids have the potential to influence cerebral amyloid aggregation, and neuroinflammation, and microbiota-associated proteopathy and neuroinflammation may be a promising area for therapeutic intervention [155]. 

Bacterial inflammagens may also indirectly contribute to the onset and progression of PD through their activation of peripheral immune cells, including macrophages, monocytes, microglia and astrocytes, which can penetrate the blood–brain barrier (BBB) and contribute to neuro-inflammation. They may also directly contribute to PD pathology by inducing structural alterations in proteins, favouring a transition from α-helices to β-sheet-rich amyloid fibrils [156]. In the brain, the resulting aggregation of these amyloid proteins leads to formation of the characteristic LBs observed in PD, and in the blood, amyloid fibrin(ogen) causes hypercoagulation, a recently discovered accompaniment of PD pathology [45]. A bacterial inflammagen that is of particular interest is bacterial cell membrane LPS, which are large molecules consisting of an inner hydrophobic lipid A domain, a non-repeating oligosaccharide ‘core’, and a distal polysaccharide chain known as an O-antigen which determine the strain’s serology [157]. The lipid A domain is typically described as the site of the molecule that is most inflammagenic [156]. We do recognise that there are commensal gut bacteria-derived LPS which may be less immunogenic [158]. However, in the context of this paper, we refer to the LPSs that can act as potent inflammagens.

### 4.2. Contribution of LPSs to Parkinson’s Disease

#### 4.2.1. LPS as a Potent Inflammagen

There are contrasting hypotheses on the effects of LPS. One such hypothesis is the hygiene hypothesis [159]. This hypothesis suggest that early exposure to specific microorganisms and parasites in infancy benefits the immune system development and confers protection against allergic and autoimmune diseases [160]. In contrast, many papers report the detrimental effects of LPS. Although LPSs and their effects are not homogenous, it should be noted that microbiome-derived LPS could possibly impact long-term immunosuppressive mechanisms in more complex ways than has been previously thought. LPS is initially extracted from bacterial membranes by serum LPS-binding protein. LPS-binding protein then transfers LPS to CD14, where CD14 then disaggregates LPS complexes and present the LPS to the toll-like receptor 4 (TLR4) [161,162]. LPS is considered a potent immune stimulator as it binds to CD14 on monocytes, and/or macrophages [161].

After LPS binds to the TLR4, multiple host cell signalling components are activated including nuclear factor-κB (NF-κB) [163,164,165,166], followed by transcription of pro-inflammatory cytokines and proteins including TNF-α, IL-1β, IL-6, IL-12, and iNOS [167,168,169,170]. Cytokines such as IL-1β and IL-6 induce the production and secretion of serum amyloid A (SAA) in the liver, specifically acute phase SAA1 and SAA2. During the acute-phase immune response, SAA contributes to the inflammatory response by attracting immune cells, activating the transcription factor NF-κB and stimulating pro-inflammatory cytokine production [171]. Intracellular LPS can also activate the noncanonical NLRP3 inflammasome pathway via caspase-11 (caspase-4 or -5 in humans), leading to caspase-1 activation [104,172].

Microglia activation is another histopathological hallmark of PD [173] and LPS can activate microglial cells, suggesting that LPS is key in the development of neuroinflammation [162,174]. LPS was also found to induce functional changes in microglia, suggesting that it will induce blood–brain barrier dysfunction due to ROS via nicotinamide adenine dinucleotide phosphate (NADPH) oxidase [175]. LPS-activated microglia can release cytokines like IL-1β, IL-6, and TNFα, resulting in an increased expression of inducible nitric oxide synthase (iNOS) and production of ROS [176]. Interestingly, it was found in a rat model that the substantia nigra had the highest density of microglia, and that these microglia were particularly vulnerable to LPS damage [173].

The damaging effects of LPS are underscored by evidence that it can cause misfolding and aggregation of α-Syn [177,178]. LPS, due to its strong inflammagenic properties, is also regularly used to produce in vivo models of both PD and Alzheimer’s disease (AD) [179,180,181], as well as other inflammatory diseases such as pre-eclampsia [182,183,184]. Systemic injection of LPS has also been shown to cause damage to the BBB of recipient animals, leading to the subsequent crossing of peripheral cytokines into the brain [185,186,187]. Low doses of LPS were also found to induce secretion of cytokines, and thereby increased vulnerability of dopamine neurons in a rat model [188].

Inflammation induced by LPS also increases α-Syn entry into the brain via the BBB [189], possibly driving LB formation (this study was an animal study). Interestingly, in a 2015 study, Hasegawa and coworkers also showed in 52 PD patients that LPS-binding protein levels were lower than in controls [103]. Lower levels of LPS-binding protein in PD might be related to its binding of increased LPS in circulation or might indicate less LPS neutralization [169]. A 2020 paper from Wijeyekoon and coworkers also directly demonstrate elevated serum endotoxin in PD, particularly in patients with increased risk for early dementia [104]. LPS in circulation, therefore, directly or indirectly leads to neurodegeneration by inducing a strong inflammatory response, causing degradation of the BBB, inflammation, and oxidative stress in the CNS and the stimulation of α-Syn misfolding and aggregation into LBs. 

#### 4.2.2. Formation of Nucleated Molecular Intermediates as Induced by LPS

It has been suggested that LPS may change the structure of (healthy) proteins, whereby it may induce the formation of nucleated molecular intermediates. Nucleation is a process whereby phase transitions are initiated in proteins [190]. In the context of protein biophysics, nucleation is used to describe a process whereby supersaturated protein solutions form insoluble macromolecular protein aggregates. Moreover, nucleation, a complicated biophysical process, can be described as a primary and a secondary process, and these processes were already described in the 1960s [191]. Nucleation therefore occurs by two pathways, the first being a fibre-independent (primary) pathway and the second a fibre-dependent (secondary) pathway [192]. Primary pathways, such as homogeneous nucleation, generate new aggregates at a rate dependent on the concentration of monomers alone and independent of the concentration of existing fibrils [191,193]. Secondary pathways are the complementary mechanisms, that generate new aggregates at a rate dependent on the concentration of existing fibrils [193]. The latter class can be subdivided into monomer-independent processes, such as fragmentation [192,193,194] with a rate depending only upon the concentration of existing fibrils, and monomer-dependent processes, such as secondary nucleation. LPS may be involved in both primary and secondary nucleation reaction pathways.

In AD, e.g., pathological protein fibril formation where protein is changed to amyloid fibrils, is well known and these fibrils are characterized by a highly ordered cross-β conformation [195]. The generation of toxic oligomers during the aggregation of the beta-amyloid (Aβ) peptide Aβ42 into amyloid fibrils and plaques is an example of a form of nucleation [193]. Cohen and coworkers showed that once a small but critical concentration of amyloid fibrils has accumulated, the toxic oligomeric species are predominantly formed from monomeric peptide molecules through a fibril-catalysed secondary nucleation reaction. The authors argued that such a secondary nucleation mechanism is seen in Aβ, rather than through a classical mechanism of homogeneous primary nucleation [193]. This catalytic mechanism, insoluble amyloid fibrils, and the generation of diffusible oligomeric aggregates are possibly the neurotoxic agents in AD. In PD, α-Syn is also a well-known example of protein conformational changes, and primary nucleation was also noted in α-Syn protein [196]. 

LPS can modulate α-Syn amyloidogenesis through the formation of intermediate nucleating species [23]. Bhattacharyya et al. demonstrated that *E. coli*-derived LPS modulates α-Syn aggregation in vitro by forming intermediate LPS-α-Syn complexes [23]. These intermediate complexes might be responsible for the PD-associated pathological effects of α-Syn amyloids [23]. The authors furthermore proposed that N-terminal-mediated anchorage of the amphipathic molecule results in the eventual partial internalization of the “fibrillating” (fibril forming) motif, situated in the hydrophobic acyl region of the LPS molecule. It, therefore, appears that LPS can modulate the overall aggregation kinetics of α-Syn in a concentration-dependent manner. Thus, the direct molecular interaction with LPS results in the modulation of the protein’s conformation into alternative nucleating forms that are morphologically and functionally distinct from the wild type α-Syn conformers. Most importantly, the characterization of the epitope of interaction in the LPS-mediated nucleation possibly allows for novel targets in the therapeutic interventions.

LPS may also induce amyloid forms of the clotting protein fibrin(ogen) during blood clotting and significantly contributes to systemic inflammation and coagulopathy [148,156,197]. Our research group has also shown that LPS may interact with the plasma protein fibrin(ogen), resulting in protein misfolding. These protein changes can be visualized using amyloid stains, e.g., thioflavin T and newer fluorescent markers, known as Amytrackers [148,156,197]. Figure 4 shows the diagram and micrographs from Bhattacharyya and coworkers [23], as well as an example from our previous work [45,148,156,197]. The intermediate LPS-α-Syn-complexes shown by Bhattacharyya and coworkers look very similar to the LPS–fibrinogen complexes. Bhattacharyya and coworkers’ 2019 results might, therefore, suggest that α-Syn fibrillation could possibly form part of a neuroimmunological response to bacterial inflammagens. 

### 4.3. Oral Microbiota Translocation in Parkinson’s Disease

Periodontitis and gingivitis have also been implicated in PD [27,198,199]. Specifically, periodontal inflammatory disease constitutes another point of entrance for bacteria in facilitating translocation. It was initially hypothesized that PD patients develop increased periodontal pathology, as a result of the progressive loss of self-care ability and fewer dental attendances [200,201]. However, Liu and coworkers reported, in a retrospective cohort study, that there is an increased risk of developing PD following chronic periodontitis [202]. Chen and workers also reported that individuals with periodontal inflammatory disease had a 1.4-fold higher chance (adjusted hazard ratio) of developing PD [203].

One of the Gram-negative bacteria that has been implicated as a causative agent in periodontitis and gingivitis is *Porphyromonas gingivalis* (*P. gingivalis*), and its inflammagens have been associated with the development of various inflammatory conditions [204,205,206,207,208]. It is mainly a bacterium from the mouth, however, after oral administration in animal studies, it may also induce gut dysbiosis and impaired gut barrier function [209], and can induce systemic inflammation [210]. In an attempt to elucidate the underlying mechanisms of how oral bacteria alter the gut microbiota, researchers performed serum metabolome analysis on mice treated with *P. gingivalis* [211]. Recipient mice showed elevated serum amino acids (alanine, glutamine, histidine, tyrosine, and phenylalanine) [211], suggesting an increase in bacterial communities, which yield these metabolites. As there is, therefore, a known association between periodontal disease and metabolic diseases, it is possible that *P. gingivalis* can affect the metabolites produced in the gut [211].

#### Gingipains as Potent Inflammagens from *P. gingivalis*

Gingipains (toxic bacterial proteases) are virulence factors produced by *P. gingivalis* [207]. Gingipains consist of Arg-gingipain (Rgp) (RgpA and RgpB) and Lys-gingipain (Kgp) and exist in both cell-associated and secreted forms, which play a central role in the virulence of this organism [212]. Gingipains were also found in the brains of AD patients and were implicated in the development of AD [207]. Recently, we reported on gingipains in the blood of PD patients, with a similar hypercoagulation effect as for LPS [45]. Gingipains may potentially also enter via the nasal cavity, from where it may potentially journey via the olfactory bulb. This process may potentially be underpinned by the proposed dual-hit mechanism, involving anterograde progression of pathology from the olfactory system into the temporal lobe, and retrograde progression to the brainstem [13,14]. These inflammagens also occur in different locations of the CNS and hence lead to different diseases. We suggest that the effects of the same inflammagen in different areas of the CNS, associated with either AD or PD pathologies, is possibly predetermined by the patient’s genetic predisposition, epigenetic changes, and cellular susceptibility. 

The data illustrating gingipains’ ability to induce hypercoagulation, as well as to infiltrate brain tissue and activate amyloid protein formation, make these bacterial proteases promising candidates for study when attempting to elucidate the aetiology and progression of PD. Figure 5 provides an overview of the proposed gingipain activities following translocation into the circulation.

## 5. Therapeutic Possibilities and Pharmaceutical Interventions

### 5.1. Iron Chelation

Several iron chelator molecules have been suggested as sequestering agents for unliganded iron [213,214]. Iron chelation molecules may also be useful in preventing ferroptosis in PD [215]. Neuromelanin has been shown to protect dopaminergic neurons from iron-induced damage, even in conditions of iron overload, by forming stable complexes with unliganded iron [216]. Youdim et al. (2004) developed brain-penetrable compounds, the VK-28 series, which are known to have iron-chelating properties similar to or even better than the well-known iron chelator desferrioxamine (Desferal). An important characteristic of an iron-chelating molecule should be its ability to inhibit monoamine oxidase (MAO). MAO generates H_2_O_2_, which interacts with ferrous iron to form reactive hydroxyl radicals via Fenton chemistry [63,217,218,219]. The Fenton reaction (Fe^2+^ + H_2_O_2_ → OH^−^ + HO•) is a nonenzymatic reaction that obeys mass action law, meaning that the rate of hydroxyl radical production is directly proportional to the amount of Fe^2+^ in the cell [220,221]. Unfortunately, desferrioxamine has relatively poor MAO inhibition [222]. In contrast, VK-28 has been shown to have good MAO inhibitory and iron-chelating properties [222]. More recently, VK-28 was also shown to be protective against iron toxicity and less toxic than deferoxamine [223]. 

Clinical trials of iron chelation in the treatment of PD have paid specific attention to deferiprone as a promising pharmaceutical intervention. Papers showing possibly promising efficacy with regard to neurodegeneration include [221,224,225]. However, little longitudinal clinical data is available that shows significant results, suggesting its efficacy in symptomatic improvement of pausing of disease progression. During a randomized trial of deferiprone administration to patients with early-stage PD, over a period of 6 months, decreased iron concentrations in the substantia nigra pars compacta (SNc) were found [213]. Suspension of treatment resulted in the restoration of the elevated iron levels, suggesting a return to the pathological iron dyshomeostasis, which might underlie PD. A phase 2 clinical trial of deferiprone by Martin-Bastida et al. (2017) [224] reported removal of excess iron concentrations in the dentate and caudate nucleus, but minimal symptomatic improvement in PD patients, noting that the trial had small numbers of patients, was limited to early PD and was only of 6 months’ duration. However, deferiprone is only bidentate and a relatively weak chelator [148], and its combination with the stronger, tridentate deferasirox may prove more effective. Iron chelation may have some mild side effects. However, deferiprone therapy was well tolerated by PD subjects with only minor side effects including exacerbation of pre-existing muscular/and joint pain or mild gastrointestinal upset [224].

The marginal success of simple metal chelation drugs like deferiprone is also possibly due to the multifactorial aetiology of PD, and the various pathological feedback loops involved in disease pathology. Simply removing one feature of the disease would do little to stop its progression or reverse its debilitating effects. Multifunctional agents, capable of targeting several underlying pathological mechanisms, have been in development for almost two decades [217,226]. Compounds like piperaxine-8-OH-quinolone hybrids have been shown to have free radical scavenging properties, independent of their iron chelation function [227]. Multifunctional iron chelators, like 7,8-dihydroxycoumarin derivative (DHC12) and coumarin-tris hybrid (CT51), have been designed to accumulate in the mitochondria, where both iron and ROS exist in high concentration, thereby increasing their antioxidant and mitochondriotropic effects [228,229]. These agents are, however, still in the experimental phase and are yet to be proven effective during clinical trials. Ergothioneine is another promising iron chelator that is also an antioxidant [230].

### 5.2. Antibiotics and Probiotics

The effective use of therapeutics such as antibiotics (and probiotics) in the treatment of PD and its associated gut-related issues can only be rigorously assessed in randomized double-blind controlled clinical trials. Unfortunately, most papers that report on the use of antibiotics and probiotics are based on limited data or case studies. However, both antibiotics and probiotics usage have been suggested for the treatment of PD, and particularly for the restoration of the gut microbiome in PD patients. Probiotics were found to possibly alter the clinical progression of PD [231] and alleviate constipation and gut-related issues [232,233]. Furthermore, probiotics, prebiotics, and synbiotics are being examined that might influence the gut–brain axis by altering gut microbiota composition, enteric nervous system, and CNS [234] and may play important roles in the regulation of dysbiosis in PD [235].

Antibiotics as treatment regime in PD have mostly focussed on targeting constipation and gut dysbiosis. Antibiotics such as rifaximin with poor systemic absorption may be used to treat small bowel bacterial overgrowth, which is also observed in PD [236]. Particularly, minocycline may have some neuroprotective activity in various experimental models including PD, [28,237,238]. Minocycline and its effects as a neuroprotecting antibiotic are related to inhibition of mitochondrial permeability and the suppression of microglial activation [239]. In addition, there is a growing body of evidence to suggest that minocycline elicits neuroprotective effects in PD, particularly because it restores gut microbiota balance, due to the reduction in Firmicutes and Bacteroidetes bacteria [25]. In addition, minocycline exerts anti-inflammatory effects that possibly may mediate its neuroprotection [240]. 

Rifampicin, another antibiotic known to exert multiple neuroprotective functions is suggested as potential treatment regime for PD [241,242]. Rifampicin and its derivative rifampicin quinone were found to reduce microglial inflammatory responses and neurodegeneration induced in vitro by α-Syn fibrillary aggregates [243]. Rifampicin might also reduce the cytotoxicity by promoting SUMOylation of α-Syn [244]. SUMOylation refers to when a small ubiquitin-like modifier (SUMO) moiety is covalently linked to a lysine residue in the target protein. Dysregulation of SUMOylation of extranuclear proteins is strongly implicated in neurological and neurodegenerative diseases [245], including PD [246]. Although there is great potential for treating PD using antibiotics, certain antibiotics, including tetracyclines, sulphonamides, and trimethoprim have been associated with increased risk of PD [247]. The exact mechanisms of action of these antibiotics in the treatment of PD is an important research question that needs to be answered. In particular, longitudinal studies are required to determine how affective antibiotics are to successfully treat the comorbidities of gut dysbiosis, gingivitis, and periodontitis, as well as constipation, or indeed to stop the progression of the disease. 

### 5.3. Faecal Microbiota Transplantation

As mentioned previously, faecal transplantation of gut microbiota from PD patients enhanced α-Syn-mediated motor dysfunction in a mouse model of PD [127]. The question that now arises is whether the opposite might be true: can faecal transplants form a healthy individual assist in the treatment of PD patients? Indeed this seems to be the case, as faecal microbiota transplantation in PD is being investigated [248]. Faecal transplantation may be important to assist in recolonizing the gut microbiome of patients with neurodegenerative diseases [249]. There has been one report of a case studies where faecal transplant were used in PD [250]. Faecal microbiota transplantation was also shown to protect mice in a PD model by suppressing neuroinflammation and reducing toll-like receptor (TLR)4/TNF-α signalling [251]. Once again, to fully understand the usefulness of this method and to present convincing clinical results, controlled clinical trials are needed. 

### 5.4. Additional Therapeutic Options 

Therapeutic interventions that directly target both LPS and gingipains could also be additional therapeutic options. Examples of such options may include small molecule inhibitors of gingipains [207]. An interesting option is the adjunctive use of lozenges containing IgY antibody against gingipains from *P. gingivalis.* It was shown that the use of this therapy resulted in clinical and microbiological benefits in the treatment for chronic periodontitis [252] and may actually also have therapeutic effects in PD. 

## 6. Conclusions

In this review, we have brought together evidence that the development and presence of PD depends on specific sets of interlinking factors that include neuro-inflammation, systemic inflammation, α-Syn-induced cell damage, vascular dysfunction, and iron dysregulation, together with gut and periodontal dysbiosis. Published evidence substantiates the view that PD is both associated with and driven by dysregulated circulating inflammagens (such as LPSs and gingipains), iron, and cytokines. Bacterial inflammagens may either enter via the gut or as a dual-hit mechanism, involving anterograde progression of pathology from the olfactory system into the temporal lobe, and retrograde progression to the brainstem [13,14], and these processes may provide areas for therapeutic intervention. There is also evidence that pre-and probiotics, as well as antibiotics and faecal transplant treatment, might be valuable treatments in PD. A current challenge for drug discovery designed for complex brain disorders such as PD is to look for multimodal drugs that might deliver disease-modifying outcomes. Targeting underlying mechanisms of PD, such as gut dysbiosis and iron toxicity, have elucidated a wide variety of novel treatments, which could not only relieve the characteristic motor deficits seen in PD but also might significantly slow the progression of the disease. We suggest that the most effective approach to prevent PD and its worsening is to determine the origin of the disease and its comorbidities and to follow a personalized treatment regime, of which we outline the main features. Most treatment options discussed here may be most effective against the comorbidities that are related to nonmotor symptoms; such symptoms may mostly precede motor symptoms. In such a personalized treatment approach, all multifactorial features should be explored. Ultimately, we need to embark on long-term longitudinal studies where large cohort data are available. 

## Figures and Tables

**Figure 1 biomolecules-11-00030-f001:**
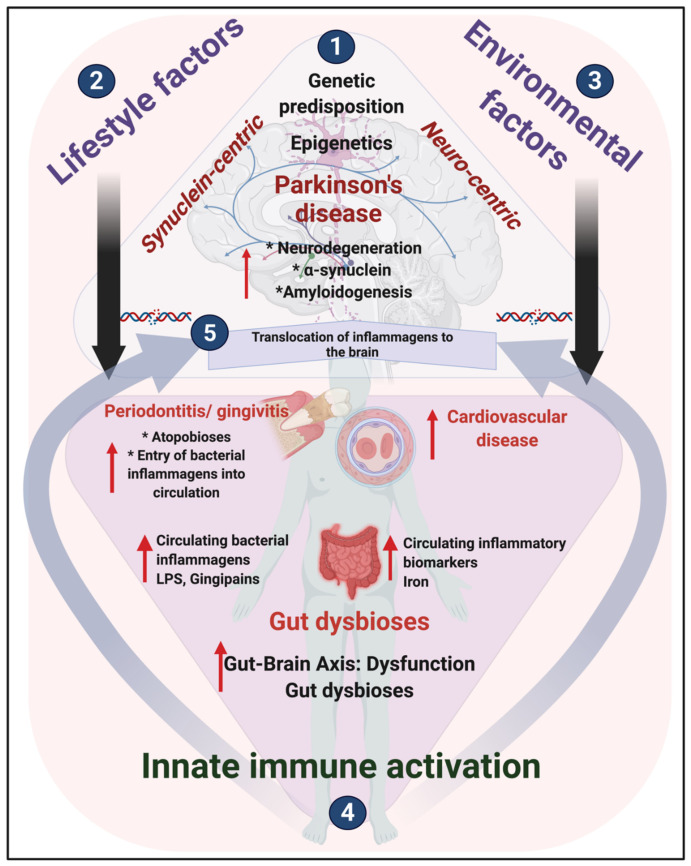
An overview of Parkinson’s disease (PD): what do we know from the literature? (1) Traditional focus on a synuclein-centric and neuro-centric- approach, where researchers predominantly focused for the origins of PD in the central nervous system (CNS). Recently, there has been a shift in focus to look closer at the role of both (2) lifestyle and (3) environmental factors that result in (4) innate immune activation, in the development of PD; these 3 factors (depicted in (2), (3) and (4), directly impact and play a significant role in PD brain neurodegeneration. (5) Inflammagens and inflammatory cytokines from the periphery can translocate to the brain.

**Figure 2 biomolecules-11-00030-f002:**
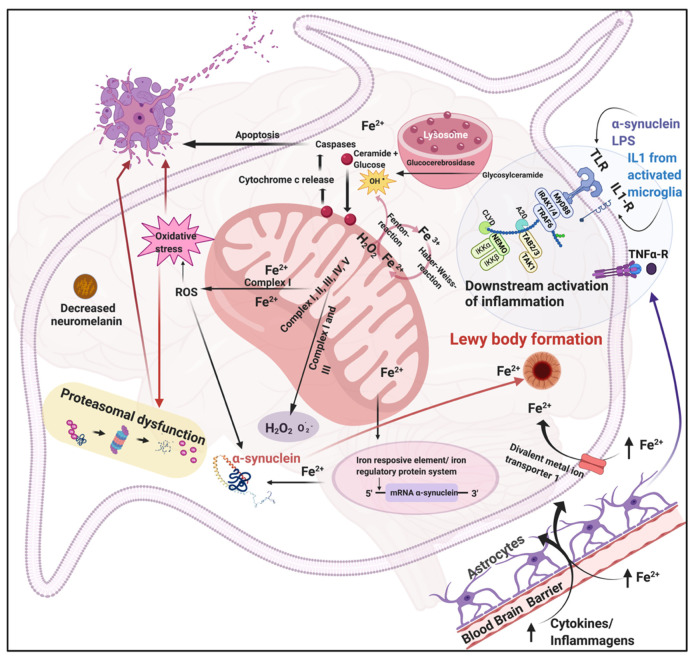
Interactions in a PD neuron between mitochondria, lysosomes, iron, and α-Syn production and ultimately proteasomal dysfunction, oxidative stress and apoptosis (adapted from [20]). The figure illustrates the interplay between mitochondria and lysosomes during the apoptotic cell death that characterizes the death of neurons in the substantia nigra that is the hallmark of PD. The location of receptors like the TLR and IL-1 receptors and the metal ion transporter 1 are shown. These receptors and transporter are known to bind bacterial inflammagens, IL-1, and Fe^2+^. In PD, there is also a loss of neuromelanin in the substantia nigra, and this could lead to enhanced calcium messaging. These molecules play a fundamental role in the downstream activation of inflammation and oxidative stress, and ultimately play a crucial role in α-Syn and Lewy body formation.

**Figure 3 biomolecules-11-00030-f003:**
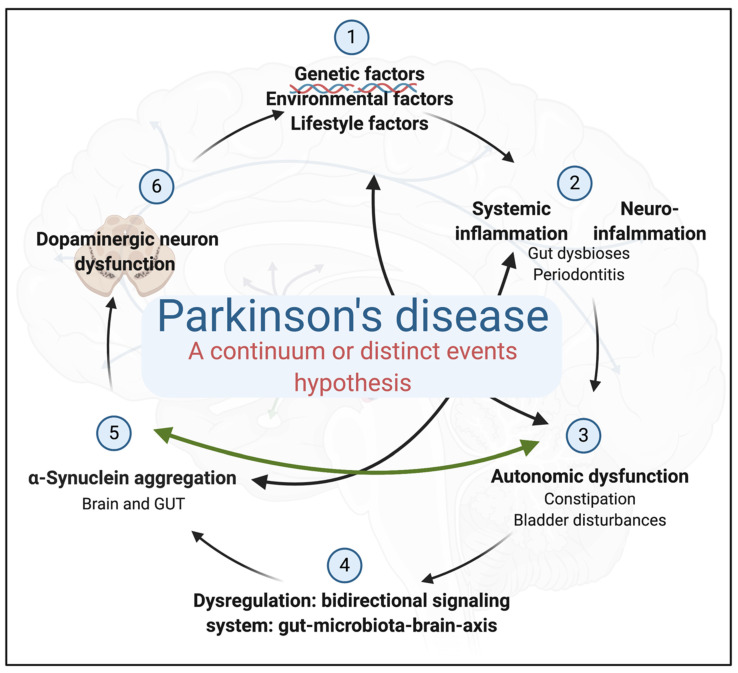
Parkinson’s disease: a continuum or distinct events hypothesis? The condition is characterized by the presence of (1) predisposing genetic, environmental, and lifestyle factors that together contribute to both (2) systemic and neuro-inflammation. (3) Autonomic dysfunction and a (4) dysregulated bidirectional signalling system and (5) α-Syn aggregation are associated with (6) motor and dopaminergic neuron dysfunction. We argue that there is a positive feedback mechanism between (5) α-Syn aggregation and (3) autonomic dysfunction.

**Figure 4 biomolecules-11-00030-f004:**
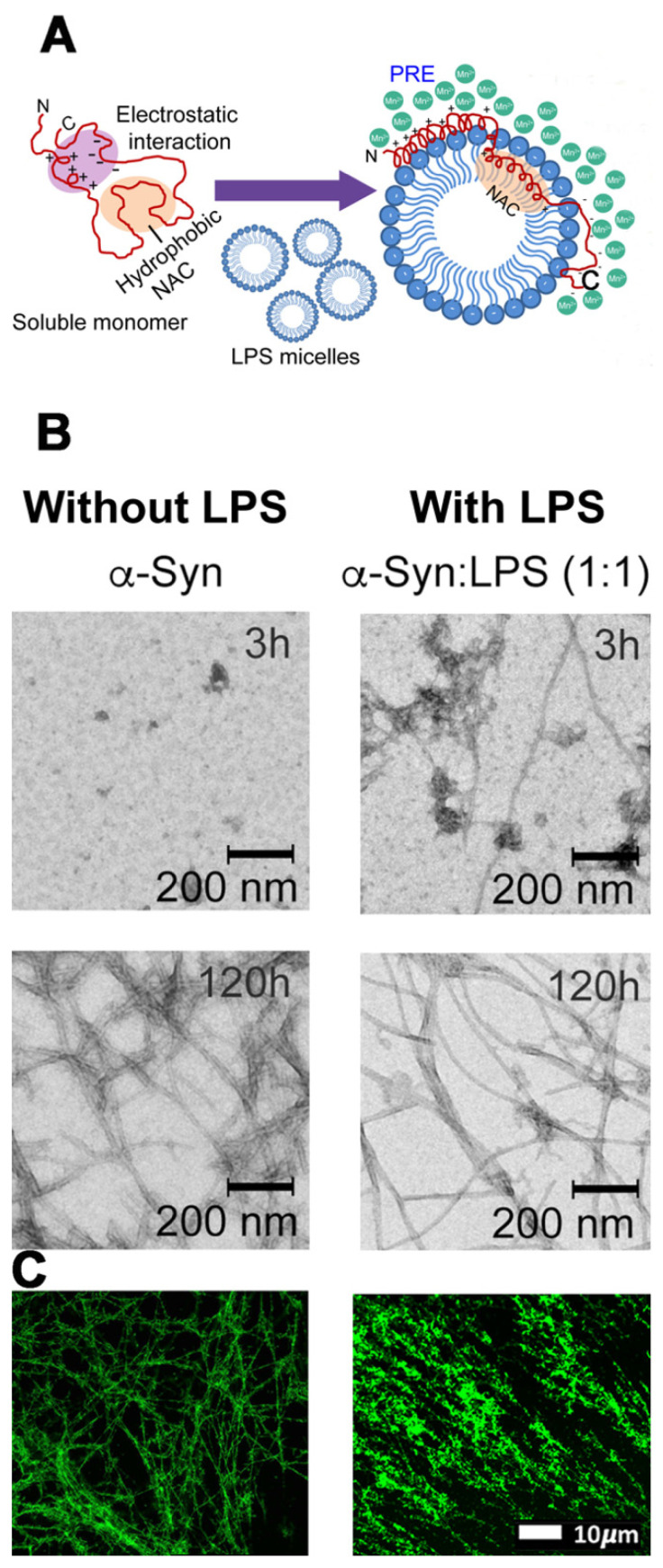
Courtesy of [23] (permission and license of usage were granted by publisher and supported by authors).(**A**) Schematic representation of the orientation of α-Syn in aqueous solution (left) and in the presence of lipopolysaccharides (LPS) micelle (right). Paramagnetic relaxation enhancement (PRE), using MnCl_2_ as quenching agent, herein indicated only as “Mn”, was used by the authors to verify internalization of the N-terminal- and NAC regions of α-Syn into the LPS micelle. (**B**) Transmission electron microscopy (TEM) images showing morphologically distinct α-Syn fibrils at 1:1 LPS concentration at two different time points (*t* = 3 and 120 h). Scale = 200 nm. (**C**) Confocal microscopy images of fibrin networks formed from purified fibrinogen (with added Alexa 488 fluorophore) incubated with and without LPS from *P. gingivalis*, followed by addition of thrombin to create extensive fibrin(ogen) clots (unused raw data from [45].

**Figure 5 biomolecules-11-00030-f005:**
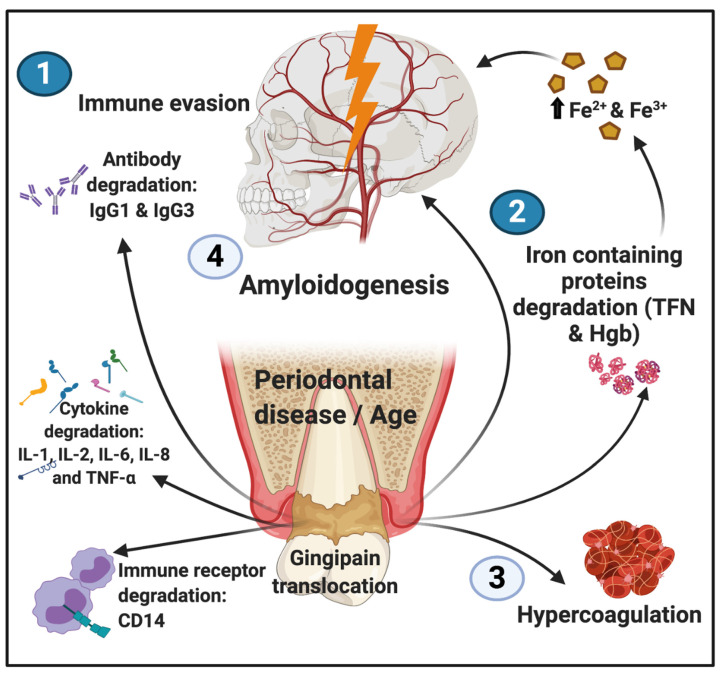
An overview of the effects of the translocation of bacterial gingipain into the circulation: (1) immune evasion brought about by proteolytic degradation of antibodies, cytokines, and immune receptor CD14; (2) proteolytic degradation of transferrin (TFN) and haemoglobin (Hgb); (3) hypercoagulation caused directly by contact of platelets with gingipain proteases; and (4) amyloidogenesis, resulting not only from direct contact of neurons with gingipain proteases but also indirectly as a result of increased iron levels, which occur due to the degradation of iron-containing proteins.

**Table 1 biomolecules-11-00030-t001:** Selected references showing evidence of systemic mediators and iron dysregulation of inflammation in Parkinson’s disease.

Mediators of Inflammation	References
Presence of activated microglia, dysregulated inflammatory mediators, chemokines, oxidative stress, and both systemic and CNS inflammation	[54,83,84,85,86,87]
Presence of dysregulated cytokines, including interleukin (IL)-1β, IL-2, IL-6, IL-10, tumour necrosis factor (TNF)-α, interferon (IFN)γ, RANTES), and C-reactive protein (CRP)	[45,88,89,90,91,92,93]
Increased cluster of differentiation (CD) 4+ T-cells indicating peripheral lymphocyte activation	[94,95]
Presence of nitric oxide synthase (iNOS) and cyclooxygenase-2 (COX2) in postmortem PD brains	[96,97]
Increase in gut–brain axis and intestinal inflammation. An increase in enteric inflammation associated with increased mRNA and mRNA that are associated with glial markers	[10,42,43,98,99]
Increased presence of stool immune factors	[100,101]
Dysregulated bacterial inflammagens like LPS (lipopolysaccharides) and bacterial proteases like gingipains	[45,102,103,104]
Iron dysregulation	[20,50,62,63,105,106,107,108,109,110,111,112,113,114,115]

## Data Availability

Not applicable.

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
