# Peer review of "Iron Dysregulation and Inflammagens Related to Oral and Gut Health Are Central to the Development of Parkinson’s Disease"

_biomolecules, 2020, doi:10.3390/biom11010030_

Round 1

Reviewer 1 Report

The article presented by van Vuuren and collaborators is a review article on the involvement of various inflammagens in the development of Parkinson Disease, focused on bacterial molecules (i.e. LPS and gingipains) as a result of oral and gut dysbiosis, and on iron dysregulation. The authors describe how these factors impact systemic inflammation, neuro-inflammation, alpha synuclein aggregation/pathology, and vascular dysfunction. Finally, they discuss new therapeutic opportunities around these inflammagens to relieve PD symptoms and/or even modify disease course.

The topic of the article is relevant and interesting for the PD field, both for the understanding of pathophysiological mechanisms and for the development of future therapeutic strategies. The text is accompanied by very illustrative and nice graphic representations.

However, there are some major aspects that should be addressed before the article is suitable for publication:

  • The title is not appropriate in the sense it includes several and very diverse concepts, it seems more of a Keywords list. However, once the text is read, the take home message, which I believe should be reflected by the title, is that inflammagens related to oral/gut dysbiosis and iron dysregulation are central to the development of PD. The word inflammagen already includes bacterial molecules like LPS. The word amyloid is latter not included in the abstract and I don’t see the need for it being in the title.
  • The abstract doesn’t really reflect what is latter explained and developed in detail along the text. I suggest the authors revise the structure of the abstract.
  • In the last sentence from the abstract, targeting gut dysbiosis and iron toxicity can directly impact on non-motor and gut-related conditions, but, couldn’t it also impact motor symptoms? Specially in terms of iron toxicity since dopaminergic cells in the substantia nigra, whose degeneration is responsible for the motor symptoms, are known to accumulate significant amounts of iron.
  • Line 66: I would suggest not to use the concept rostro-caudal since in the case of the brainstem pathology, a-syn deposition follows a retrograde progression from the dorsal motor nucleus of the vagus to more rostral brain regions.
  • Towards the end of the introduction, the two aspects that the authors point as involved in the crucial systemic inflammation seen in PD (line 87), which are bacterial molecules and iron, do not seem to have an equal importance. “Iron” seems to disappear and leave all the space to the microbiome-gut-brain axis (example line 102 and 107). Since both topics, bacteria and iron, seem to be the focus of the paper according to the title, I suggest balancing both aspects a bit more in the introduction and give more specific weight to iron dysregulation, because otherwise it seems a review essentially focused on gut dysbiosis and its consequences on inflammation. Also, both concepts should be better balanced along the text since only half-page is dedicated to iron while the rest of the review is dedicated to bacterial inflammagens.
  • Section 2 is lacking an introductory paragraph and a conclusive sentence. Also, a sub-section 2.1 is made, while there is no sub-section 2.2. It could probably be then just all considered as section 2. The title given to section 2 does not correspond to the content of the section.
  • Please better develop the concept of pathological levels of iron and oxidative stress and importance to PD neuro-inflammation. No experimental evidences are directly referred to in order to support this link, but the references given are directly related to iron accumulation, not its effects at the level of inflammation.
  • Figure 1 depicts iron next to circulating inflammatory biomarkers but this peripheral role of iron is not described nor further developed in section 2 when talking about iron. Why is inflammation and iron accumulation only represented in the periphery and not also added next to the brain mechanisms (i.e. neurodegeneration, synuclein, amyloidogenesis)?
  • The second paragraph in section 4 refers to the hypothesis that alpha-syn aggregation originates in the gut and propagates then to the brain. This is not something suggested by the authors, but a hypothesis that has been on the table for many years. Thus, I would suggest to remove “we suggest” from line 335.
  • The sentence 341 “An important consideration …” seems to be in the middle of two sentences that should be consecutive since they are referring to animal models of a-syn propagation.
  • Please describe better which “bacterial amyloids” line 361 refers to.
  • Section 4.2.1 contains a lot of information and sentences are not well ordered, please revise.
  • Line 463 in between parenthesis: this is an interesting concept, but should be better explained since it is somehow opposite to the fact that inflammation caused by bacteria can exacerbate synuclein aggregation.
  • Line 469: “LPS is also implicated in protein nucleation” is redundant, has been previously mentioned and is not necessary to introduce the paragraph.
  • Section 4.3 refers to the Oral microbiota, while the title of Section 4 refered to Gut Dysbiosis. Please change the title in section 4 so that it includes gut and oral dysbiosis.
  • Rephrase the last conclusion sentence in section 4.3, it is not clear.
  • Line 534 “We suggest that the presence …. hence to different diseases.” doesn’t fit with previous and next sentences. Also, if the inflammagens enter the brain via systemic circulation and/or BBB leakage, how do they target different neuronal populations in both pathologies?
  • The order of section 5 is not logical. Section 5.2 about iron chelation should be the first one since iron dysfunction was discussed first in the text and since, in this way, all modulations related to the microbiota, antibiotics and probiotics and fecal transplants, are explained in a consecutive manner.
  • Conclusion line 664 “In addition, bacterial inflammagens such as LPS, …, may also fuel inflammation …”. Aren’t bacterial inflammagens already included in the “dysregulated circulating inflammagens” refered to in the previous sentence?

Minor points:

  • The order of some sentences is confusing and does not always follow a logical order. For example, line 54 “Neuronal lesions …” seems more of a conclusion sentence at the end of this paragraph (line 63).
  • Figure 1: Why is amyloidogenesis in capital letters while the rest are not?
  • Figure 2: integrate neuromelanin as an important iron chelator in your picture.
  • Table 1 contains references regarding systemic inflammatory mediators and iron dysfunction in PD, although the title of the table does not include the iron part.
  • The sentences selected in Table 1 to summarize the key concepts of the related references are not really understandable, please reformulate.
  • References are missing in the sentence line 233-235
  • Please revisit grammar in sentence line 254 “it may be worth …”.
  • The last sentence in section 3 (line 279) is not a conclusion for section 3 but an introduction for section 4, which is not really needed here since the concept has already been introduced before.
  • LPS is used in Table 1 and line 373 without being abbreviated (not done until section 4.2.1)
  • Line 428: LPS binding protein was previously abbreviated as LBP
  • Line 444: fibre or fiber-dependent?
  • Figure 4: Comparing images between panel B and C (specially the right panels) doesn’t really show that the complexes look very similar. Do you have better pictures?
  • Figure 5: I would suggest changing “TFN and Hgb degradation” to “Iron containing proteins degradation (TFN and Hgb)” to be more consistent with the other points and to make the figure more self-explanatory.
  • Line 646: “There has been reports …”. Only one reference is given, it should be changed to “One report …”.

Author Response

Dear Reviewer 1, please find our amendments.  In the uploaded paper, the amendments that you suggested, are shown in red.

  • The title is not appropriate in the sense it includes several and very diverse concepts, it seems more of a Keywords list. However, once the text is read, the take home message, which I believe should be reflected by the title, is that inflammagens related to oral/gut dysbiosis and iron dysregulation are central to the development of PD. The word inflammagen already includes bacterial molecules like LPS. The word amyloid is latter not included in the abstract and I don’t see the need for it being in the title.

We have changed the title to:

Inflammagens related to oral and gut health are central to the development of Parkinson’s disease

  • The abstract doesn’t really reflect what is latter explained and developed in detail along the text. I suggest the authors revise the structure of the abstract.

The abstract was adapted

  • In the last sentence from the abstract, targeting gut dysbiosis and iron toxicity can directly impact on non-motor and gut-related conditions, but, couldn’t it also impact motor symptoms? Specially in terms of iron toxicity since dopaminergic cells in the substantia nigra, whose degeneration is responsible for the motor symptoms, are known to accumulate significant amounts of iron.

Thank you, we agree that we should also add motor symptoms into this sentence, it now reads:

However, targeting underlying mechanisms of PD, including gut dysbiosis and iron toxicity, have potentially opened up possibilities of a wide variety of novel treatments which may relieve the characteristic motor and non-motor deficits of PD, and may even slow the progression and/or accompanying gut-related conditions of the disease.

  • Line 66: I would suggest not to use the concept rostro-caudal since in the case of the brainstem pathology, a-syn deposition follows a retrograde progression from the dorsal motor nucleus of the vagus to more rostral brain regions.

Apologies is should be caudo-rostral.

  • Towards the end of the introduction, the two aspects that the authors point as involved in the crucial systemic inflammation seen in PD (line 87), which are bacterial molecules and iron, do not seem to have an equal importance. “Iron” seems to disappear and leave all the space to the microbiome-gut-brain axis (example line 102 and 107). Since both topics, bacteria and iron, seem to be the focus of the paper according to the title, I suggest balancing both aspects a bit more in the introduction and give more specific weight to iron dysregulation, because otherwise it seems a review essentially focused on gut dysbiosis and its consequences on inflammation. Also, both concepts should be better balanced along the text since only half-page is dedicated to iron while the rest of the review is dedicated to bacterial inflammagens.

Perhaps the title change now places less emphasis on iron specifically, addressing the weight issue of the various concepts?  Your suggestion with regards to section 2 headings, below, also perhaps ring-fences iron into a clear focussed section.

  • Section 2 is lacking an introductory paragraph and a conclusive sentence. Also, a sub-section 2.1 is made, while there is no sub-section 2.2. It could probably be then just all considered as section 2. The title given to section 2 does not correspond to the content of the section.

Thank you for this suggestion, we deleted the original title for the section:

  1. A mind-shift in Parkinson’s disease research

And changed 2.1 to the new section heading, so 2. Now reads

  1. Parkinson’s disease, iron, oxidative stress and chronic systemic inflammation
  • Please better develop the concept of pathological levels of iron and oxidative stress and importance to PD neuro-inflammation. No experimental evidences are directly referred to in order to support this link, but the references given are directly related to iron accumulation, not its effects at the level of inflammation.

We have added the following paragraph:

…Pathological levels of iron and oxidative stress are also central to the persistence of neuro-inflammation.  A cycle of decreased levels of endogenous antioxidants, increased ROS, augmented dopamine oxidation, and high iron levels have been found in brains from PD patients (Sutachan et al., 2012).  Examples are decreased presence Coenzyme-Q10 (CoQ10), uric acid and vitamin E (Chang and Chen, 2020).   CoQ10 scavenges free radicals, with main function of protection of mitochondrial and lipid membranes, including (Sohmiya et al., 2004).  Uric acid, which acts as an antioxidant, is also lower in PD (Sohmiya et al., 2004).  Vitamin E is known to prevent lipids from oxidative stress, and plasma levels ov vitamin E are reduced in PD patients (Nicoletti et al., 2001).  Dopamine metabolism, high levels of iron and calcium in the substantia nigra, mitochondrial dysfunction and neuroinflammation directly contribute to the increased oxidative stress and dopanergic neuronal loss in the brains of PD patients  (Chang and Chen, 2020, Mochizuki et al., 2020).  In a 2016 study, serum levels  including that of iron, ferritin, transferrin, superoxide dismutase catalase, nitrosative stress marker, thiobarbituric acid reactive substances,  and other similar oxidative stress markers were analysed in 40 PD patients and 46 controls (Medeiros et al., 2016).  The authors concluded that ROS/RNS production and neuroinflammation may dysregulate iron homeostasis and that oxidative stress may be key drivers in the pathophysiology of PD. See Table 1 for selected references regarding systemic inflammatory mediators and numerous papers reporting on a dysfunction of iron metabolism in PD.

  • Figure 1 depicts iron next to circulating inflammatory biomarkers but this peripheral role of iron is not described nor further developed in section 2 when talking about iron.

We have added the following paragraph early in section 2:

…The spectrum of iron dysregulation in PD includes iron storage, uptake and release (Jiang et al., 2017); and importantly, the effects may be seen both in the brain and also the peripheral tissues.   Iron dysregulation in PD, occurs both in the brain and in circulation (Figure 1) (for a review see (Pretorius et al., 2014)).  It was previously shown that increased serum ferritin levels are present in PD, where it may directly cause eryptosis of erythrocytes (Pretorius et al., 2014).  In circulation, iron dysregulation is directly implicated as a major cause of oxidative stress (Kell and Pretorius, 2018, Galaris et al., 2019).  Furthermore, circulating iron and serum ferritin dysregulation  (due to the release of poorly liganded iron) causes pathological changes in both erythrocyte and fibrin morphology(Pretorius et al., 2013a, Pretorius et al., 2013b, Kell and Pretorius, 2014, Pretorius and Kell, 2014).  In 2012, it was also reported that serum ferritin levels were significantly increased in male and female PD patients, and were correlated with PD severity stages and duration in men and women (Madenci et al., 2012).   

  • Why is inflammation and iron accumulation only represented in the periphery and not also added next to the brain mechanisms (i.e. neurodegeneration, synuclein, amyloidogenesis)?

Apologies the two grey arrows was supposed to suggest that iron/inflammagens/biomarkers travel translocate to the brain.  The Figure was adjusted to clarify.

  • The second paragraph in section 4 refers to the hypothesis that alpha-syn aggregation originates in the gut and propagates then to the brain. This is not something suggested by the authors, but a hypothesis that has been on the table for many years. Thus, I would suggest to remove “we suggest” from line 335.

removed

  • The sentence 341 “An important consideration …” seems to be in the middle of two sentences that should be consecutive since they are referring to animal models of a-syn propagation.

The words “An important consideration” were removed and the sentences restructured as follows:

When gut dysbiosis is present in PD patients, gut-derived bacteria escape from the gastrointestinal tract into the blood.  This process leads to shedding of endotoxins into the systemic circulation, which may constitute a trigger event in the development of PD and other neurodegenerative disorders (Adams et al., 2019, Clarke et al., 2013).  

  • Please describe better which “bacterial amyloids” line 361 refers to.

The sentences now read:

Many bacteria can also assemble functional amyloid fibers on their cell surface and these amyloids contribute to biofilm formation where cells interact with a surface or with other cells (Evans et al., 2018).  Furthermore, these bacterial amyloids  have the potential to influence cerebral amyloid aggregation, and neuroinflammation, and  microbiota-associated proteopathy and neuroinflammation may be a promising area for therapeutic intervention (Friedland and Chapman, 2017). 

  • Section 4.2.1 contains a lot of information and sentences are not well ordered, please revise.

This section was revised and sentences re-ordered for better flow between concepts.

  • Line 463 in between parenthesis: this is an interesting concept, but should be better explained since it is somehow opposite to the fact that inflammation caused by bacteria can exacerbate synuclein aggregation.

We have removed the following paragraph as it is not within the scope of the current paper

(Although not the focus of this review, we also note that both Aβ and α-Syn have antimicrobial properties and there are experimental evidence for this happening, as shown in mouse (Tomlinson et al., 2017) and worm models (Kumar et al., 2016).    There is also evidence that Aβ oligomers have potent, broad-spectrum antimicrobial properties by forming fibrils that entrap pathogens and disrupt cell membranes (Kumar et al., 2016). Furthermore, it was also shown that α-Syn exhibits antibacterial activity against E. coli and Staphylococcus aureus (Park et al., 2016).)

 removed

  • Line 469: “LPS is also implicated in protein nucleation” is redundant, has been previously mentioned and is not necessary to introduce the paragraph.

removed

  • Section 4.3 refers to the Oral microbiota, while the title of Section 4 refered to Gut Dysbiosis. Please change the title in section 4 so that it includes gut and oral dysbiosis.

changed

  • Rephrase the last conclusion sentence in section 4.3, it is not clear.

Changed to now read:

As there is therefore a known association between periodontal disease and metabolic diseases, it is possible that P. gingivalis can affect the metabolites (Kato et al., 2018).

  • Line 534 “We suggest that the presence …. hence to different diseases.” doesn’t fit with previous and next sentences. Also, if the inflammagens enter the brain via systemic circulation and/or BBB leakage, how do they target different neuronal populations in both pathologies?

We moved the sentence lower and changed it to now read:

We suggest that the effects of the same inflammagen in different areas of the CNS,  associated with either AD or PD pathologies, is possibly predetermined by the patient’s genetic predisposition, epigenetic changes and cellular susceptibility.

The order of section 5 is not logical. Section 5.2 about iron chelation should be the first one since iron dysfunction was discussed first in the text and since, in this way, all modulations related to the microbiota, antibiotics and probiotics and fecal transplants, are explained in a consecutive manner.

Thank you, this makes better sense

  • Conclusion line 664 “In addition, bacterial inflammagens such as LPS, …, may also fuel inflammation …”. Aren’t bacterial inflammagens already included in the “dysregulated circulating inflammagens” refered to in the previous sentence?

Sentence removed and the one above changes slightly to now read

 Published evidence substantiates the view that PD is both associated with and driven by dysregulated circulating inflammagens (such as LPSs, and gingipains), iron and cytokines

Minor points:

  • The order of some sentences is confusing and does not always follow a logical order. For example, line 54 “Neuronal lesions …” seems more of a conclusion sentence at the end of this paragraph (line 63).

Moved

  • Figure 1: Why is amyloidogenesis in capital letters while the rest are not?

Changed

  • Figure 2: integrate neuromelanin as an important iron chelator in your picture.

Added

  • Figure 2: integrate neuromelanin as an important iron chelator in your picture.

Added

  • Table 1 contains references regarding systemic inflammatory mediators and iron dysfunction in PD, although the title of the table does not include the iron part.

Changed

  • The sentences selected in Table 1 to summarize the key concepts of the related references are not really understandable, please reformulate.

addressed

  • References are missing in the sentence line 233-235

Sentence now reads:

The etiology of PD is a complex process and the evidence in persons with PD (with or without REM behavioural disorder) indicates that degeneration may start either in the CNS or in the peripheral nervous system (PNS) (Pretorius et al., 2014), affecting numerous fundamental cellular processes (Kalia and Lang, 2015).

  • Please revisit grammar in sentence line 254 “it may be worth …”.

Sentence now reads:

It may be worth noting here that individuals without PD, may also have intestinal α-Syn inclusions, which can also be detected (Corbillé et al., 2017).

  • The last sentence in section 3 (line 279) is not a conclusion for section 3 but an introduction for section 4, which is not really needed here since the concept has already been introduced before.

The sentence was moved to section 4

  • LPS is used in Table 1 and line 373 without being abbreviated (not done until section 4.2.1)

Written out in Table 1 – first reference to it and also in abstract

  • Line 428: LPS binding protein was previously abbreviated as LBP

We rather kept the term: LPS-binding protein everywhere

  • Line 444: fibre or fiber-dependent?

Changed all to fiber-dependent (according to the original paper)

  • Figure 4: Comparing images between panel B and C (specially the right panels) doesn’t really show that the complexes look very similar. Do you have better pictures?

Unfortunately not – these figs were provided by the authors Bhattacharyya et al

  • Figure 5: I would suggest changing “TFN and Hgb degradation” to “Iron containing proteins degradation (TFN and Hgb)” to be more consistent with the other points and to make the figure more self-explanatory.

Changed

  • Line 646: “There has been reports …”. Only one reference is given, it should be changed to “One report …”.

Changed to read:

…There has been one report of a case study….

REFERENCES ADDED (REVIEWER 1 AND 2 SUGGESTIONS)

ADAMS, B., NUNES, J. M., PAGE, M. J., ROBERTS, T., CARR, J., NELL, T. A., KELL, D. B. & PRETORIUS, E. 2019. Parkinson's Disease: A Systemic Inflammatory Disease Accompanied by Bacterial Inflammagens. Front Aging Neurosci, 11,210.

CAO, J. Y. & DIXON, S. J. 2016. Mechanisms of ferroptosis. Cell Mol Life Sci, 73, 2195-209.

CHANG, K. H. & CHEN, C. M. 2020. The Role of Oxidative Stress in Parkinson's Disease. Antioxidants (Basel), 9.

CLARKE, G., GRENHAM, S., SCULLY, P., FITZGERALD, P., MOLONEY, R. D., SHANAHAN, F., DINAN, T. G. & CRYAN, J. F. 2013. The microbiome-gut-brain axis during early life regulates the hippocampal serotonergic system in a sex-dependent manner. Mol Psychiatry, 18, 666-73.

CORBILLÉ, A. G., PRETERRE, C., ROLLI-DERKINDEREN, M., CORON, E., NEUNLIST, M., LEBOUVIER, T. & DERKINDEREN, P. 2017. Biochemical analysis of α-synuclein extracted from control and Parkinson's disease colonic biopsies. Neurosci Lett, 641, 81-86.

DO VAN, B., GOUEL, F., JONNEAUX, A., TIMMERMAN, K., GELÉ, P., PÉTRAULT, M., BASTIDE, M., LALOUX, C., MOREAU, C., BORDET, R., DEVOS, D. & DEVEDJIAN, J. C. 2016. Ferroptosis, a newly characterized form of cell death in Parkinson's disease that is regulated by PKC. Neurobiol Dis, 94, 169-78.

EVANS, M. L., GICHANA, E., ZHOU, Y. & CHAPMAN, M. R. 2018. Bacterial Amyloids. Methods Mol Biol, 1779, 267-288.

FRIEDLAND, R. P. & CHAPMAN, M. R. 2017. The role of microbial amyloid in neurodegeneration. PLoS Pathog, 13,e1006654.

GALARIS, D., BARBOUTI, A. & PANTOPOULOS, K. 2019. Iron homeostasis and oxidative stress: An intimate relationship. Biochim Biophys Acta Mol Cell Res, 1866, 118535.

GUINEY, S. J., ADLARD, P. A., BUSH, A. I., FINKELSTEIN, D. I. & AYTON, S. 2017. Ferroptosis and cell death mechanisms in Parkinson's disease. Neurochem Int, 104, 34-48.

JIANG, H., WANG, J., ROGERS, J. & XIE, J. 2017. Brain Iron Metabolism Dysfunction in Parkinson's Disease. Mol Neurobiol, 54, 3078-3101.

KALIA, L. V. & LANG, A. E. 2015. Parkinson's disease. Lancet, 386, 896-912.

KATO, T., YAMAZAKI, K., NAKAJIMA, M., DATE, Y., KIKUCHI, J., HASE, K., OHNO, H. & YAMAZAKI, K. 2018. Oral Administration of Porphyromonas gingivalis Alters the Gut Microbiome and Serum Metabolome. mSphere, 3,e00460-18.

KELL, D. B. & PRETORIUS, E. 2014. Serum ferritin is an important inflammatory disease marker, as it is mainly a leakage product from damaged cells. Metallomics, 6, 748-73.

KELL, D. B. & PRETORIUS, E. 2018. No effects without causes: the Iron Dysregulation and Dormant Microbes hypothesis for chronic, inflammatory diseases. Biological Reviews, 93, 1518-1557.

KUMAR, D. K., CHOI, S. H., WASHICOSKY, K. J., EIMER, W. A., TUCKER, S., GHOFRANI, J., LEFKOWITZ, A., MCCOLL, G., GOLDSTEIN, L. E., TANZI, R. E. & MOIR, R. D. 2016. Amyloid-β peptide protects against microbial infection in mouse and worm models of Alzheimer's disease. Sci Transl Med, 8, 340ra72.

MADENCI, G., BILEN, S., ARLI, B., SAKA, M. & AK, F. 2012. Serum iron, vitamin B12 and folic acid levels in Parkinson's disease. Neurochem Res, 37, 1436-41.

MASALDAN, S., BUSH, A. I., DEVOS, D., ROLLAND, A. S. & MOREAU, C. 2019. Striking while the iron is hot: Iron metabolism and ferroptosis in neurodegeneration. Free Radic Biol Med, 133, 221-233.

MEDEIROS, M. S., SCHUMACHER-SCHUH, A., CARDOSO, A. M., BOCHI, G. V., BALDISSARELLI, J., KEGLER, A., SANTANA, D., CHAVES, C. M., SCHETINGER, M. R., MORESCO, R. N., RIEDER, C. R. & FIGHERA, M. R. 2016. Iron and Oxidative Stress in Parkinson's Disease: An Observational Study of Injury Biomarkers. PLoS One, 11, e0146129.

MOCHIZUKI, H., CHOONG, C. J. & BABA, K. 2020. Parkinson's disease and iron. J Neural Transm (Vienna), 127, 181-187.

NICOLETTI, G., CRESCIBENE, L., SCORNAIENCHI, M., BASTONE, L., BAGALÀ, A., NAPOLI, I. D., CARACCIOLO, M. & QUATTRONE, A. 2001. Plasma levels of vitamin E in Parkinson's disease. Arch Gerontol Geriatr, 33, 7-12.

PARK, S. C., MOON, J. C., SHIN, S. Y., SON, H., JUNG, Y. J., KIM, N. H., KIM, Y. M., JANG, M. K. & LEE, J. R. 2016. Functional characterization of alpha-synuclein protein with antimicrobial activity. Biochem Biophys Res Commun, 478, 924-8.

PLUM, S., STEINBACH, S., ATTEMS, J., KEERS, S., RIEDERER, P., GERLACH, M., MAY, C. & MARCUS, K. 2016. Proteomic characterization of neuromelanin granules isolated from human substantia nigra by laser-microdissection. Sci Rep, 6, 37139.

PRETORIUS, E., BESTER, J., VERMEULEN, N. & LIPINSKI, B. 2013a. Oxidation inhibits iron-induced blood coagulation. Curr Drug Targets, 14, 13-9.

PRETORIUS, E. & KELL, D. B. 2014. Diagnostic morphology: biophysical indicators for iron-driven inflammatory diseases. Integr Biol (Camb), 6, 486-510.

PRETORIUS, E., SWANEPOEL, A. C., BUYS, A. V., VERMEULEN, N., DUIM, W. & KELL, D. B. 2014. Eryptosis as a marker of Parkinson's disease. Aging (Albany NY), 6, 788.

PRETORIUS, E., VERMEULEN, N., BESTER, J., LIPINSKI, B. & KELL, D. B. 2013b. A novel method for assessing the role of iron and its functional chelation in fibrin fibril formation: the use of scanning electron microscopy. Toxicol Mech Methods, 23, 352-9.

QIU, Y., CAO, Y., CAO, W., JIA, Y. & LU, N. 2020. The Application of Ferroptosis in Diseases. Pharmacol Res, 159, 104919.

SOHMIYA, M., TANAKA, M., TAK, N. W., YANAGISAWA, M., TANINO, Y., SUZUKI, Y., OKAMOTO, K. & YAMAMOTO, Y. 2004. Redox status of plasma coenzyme Q10 indicates elevated systemic oxidative stress in Parkinson's disease. J Neurol Sci, 223, 161-6.

SUTACHAN, J. J., CASAS, Z., ALBARRACIN, S. L., STAB, B. R., 2ND, SAMUDIO, I., GONZALEZ, J., MORALES, L. & BARRETO, G. E. 2012. Cellular and molecular mechanisms of antioxidants in Parkinson's disease. Nutr Neurosci, 15, 120-6.

TOMLINSON, J. J., SHUTINOSKI, B., DONG, L., MENG, F., ELLEITHY, D., LENGACHER, N. A., NGUYEN, A. P., CRON, G. O., JIANG, Q., ROBERSON, E. D., NUSSBAUM, R. L., MAJBOUR, N. K., EL-AGNAF, O. M., BENNETT, S. A., LAGACE, D. C., WOULFE, J. M., SAD, S., BROWN, E. G. & SCHLOSSMACHER, M. G. 2017. Holocranohistochemistry enables the visualization of α-synuclein expression in the murine olfactory system and discovery of its systemic anti-microbial effects. J Neural Transm (Vienna), 124, 721-738.

WEILAND, A., WANG, Y., WU, W., LAN, X., HAN, X., LI, Q. & WANG, J. 2019. Ferroptosis and Its Role in Diverse Brain Diseases. Mol Neurobiol, 56, 4880-4893.

YU, H., GUO, P., XIE, X., WANG, Y. & CHEN, G. 2017. Ferroptosis, a new form of cell death, and its relationships with tumourous diseases. J Cell Mol Med, 21, 648-657.

Reviewer 2 Report

This review manuscript by Vuuren provides the association between the pathomechanisms of Parkinson's disease and neuroinflammation from systemic inflammation due to oral and gut bacteria. The discussion of this review article is comprehensive and very helpful for understanding the pathomechanisms of Parkinson's disease focused on the inflammation linked with oral and gut microbiota. Thus, I am very interested in this article. However, I raised some comments to improve better than this version.

  1. In page 2, line 81, “mutations in these six genes explain…” I could not find the corresponding six genes.
  2. In Figure 1, the legend should be addressed more detailed explanation of this figure. Readers will be hard to understand the collaboration among 1)-4). What do the arrows indicate? What is synuclein-centeric? What is Neuro-centeric?
  3. In Figure 2, What are the cells on the blood-brain barrier? I think that they are astrocytes. The authors should address the names of the cells. In addition, neuromelanin should be explained in this figure.
  4. In Figure 3, The authors described “6) motor and dopaminergic neuron dysfunction. However, motor neurons usually are not included in the pathomechanisms of Parkinson’s disease.
  5. The authors should explain Figure 4A. What are PRE and Mn? I think Mn will indicate manganese. But the authors did not explain the role of manganese in the aggregation of alpha-synuclein induced by lipopolysaccharide.
  6. Page 16, the authors should describe the association between ferroptosis and neurodegeneration in Parkinson's disease

Author Response

Dear Reviewer 2, thank you for your recommendations.  We have amended the paper, please find your amendments in green in the uploaded paper.

  1. In page 2, line 81, “mutations in these six genes explain…” I could not find the corresponding six genes.

In the following paper (our reference 21), the following is stated, and we therefore refer to this statement in the paper:

“Just how complex it is, is underlined by the notion that today, nearly 15 years later, we know of 28 distinct chromosomal regions more or less convincingly related to PD. Only six of these specific regions contain genes with mutations that conclusively cause monogenic PD; that is, a form of the disease for which a mutation in a single gene is sufficient to cause the phenotype. Even collectively, mutations in these six genes explain only a limited number (3%–5%) of sporadic disease occurrences” 

  1. In Figure 1, the legend should be addressed more detailed explanation of this figure. Readers will be hard to understand the collaboration among 1)-4). What do the arrows indicate? What is synuclein-centeric? What is Neuro-centeric?

Legend was adapted to explain terms better, it now reads:

Figure 1: An overview of Parkinson’s Disease (PD): what do we know from the literature? 1) Traditional focus on a synuclein-centric and neuro-centric- approach, where researchers predominantly focused for the origins of PD in the central nervous system (CNS). Recently, there has been a shift in focus to look closer at the role of both 2)lifestyle and 3) environmental factors that result in 4) innate immune activation, in the development of PD; these 3 factors (depicted in 2) 3) and 4), directly impact and play a significant role in PD brain neurodegeneration.  5)Inflammagens and  inflammatory cytokines from the  periphery can translocate to the brain.

2. In Figure 2, What are the cells on the blood-brain barrier? I think that they are astrocytes. The authors should address the names of the cells. In addition, neuromelanin should be explained in this figure.

Astrocytes are now named

Loss of neuromelanin indicated in Figure 2, and in legend: In PD there is also a loss of neuromelanin in the substantia nigra, and this could lead to enhanced calcium messaging.

We added sentence (lone 175): Neuromelanin is a complex polymer pigment found primarily in the dopaminergic neurons of human substantia nigra, and it is stored in granules including a protein matrix and lipid droplets (Plum et al., 2016).

3. In Figure 3, The authors described “6) motor and dopaminergic neuron dysfunction. However, motor neurons usually are not included in the pathomechanisms of Parkinson’s disease.

“Motor” was removed

4. The authors should explain Figure 4A. What are PRE and Mn? I think Mn will indicate manganese. But the authors did not explain the role of manganese in the aggregation of alpha-synuclein induced by lipopolysaccharide.

The following sentence was added to the Figure legend: Paramagnetic relaxation enhancement (PRE), using MnCl2as quenching agent, herein indicated only as ‘Mn’, was used by the authors to verify internalisation of the N-terminal- and NAC regions of α-Syn into the LPS micelle.

5. Page 16, the authors should describe the association between ferroptosis and neurodegeneration in Parkinson's disease

We have added the following paragraph in section 2. Parkinson’s disease, iron, oxidative stress and chronic systemic inflammation

Recently, ferroptosis (a kind of regulated cell death that is characterized by highly iron-dependent lipid peroxidation) has been associated with neurodegenerative conditions (Qiu et al., 2020, Weiland et al., 2019) and particularly PD (Do Van et al., 2016).  Ferroptosis happens due to the depletion of plasma membrane unsaturated fatty acids and accumulation of iron-induced lipid ROS (Cao and Dixon, 2016).  The over-accumulation of lipid ROS leads to an oxidative stress response in cells that causes lethal damage to proteins, nucleic acids, and lipids and eventually to cell death (Yu et al., 2017, Weiland et al., 2019). For a comprehensive review on ferroptosis in PD, see (Guiney et al., 2017).

We have added the following paragraph in section 5.1 (iron chelation section)

Iron chelation molecules may also be useful in preventing ferroptosis in PD (Masaldan et al., 2019). 

References that was added/ used in the paper in the amended sections

ADAMS, B., NUNES, J. M., PAGE, M. J., ROBERTS, T., CARR, J., NELL, T. A., KELL, D. B. & PRETORIUS, E. 2019. Parkinson's Disease: A Systemic Inflammatory Disease Accompanied by Bacterial Inflammagens. Front Aging Neurosci, 11,210.

CAO, J. Y. & DIXON, S. J. 2016. Mechanisms of ferroptosis. Cell Mol Life Sci, 73, 2195-209.

CHANG, K. H. & CHEN, C. M. 2020. The Role of Oxidative Stress in Parkinson's Disease. Antioxidants (Basel), 9.

CLARKE, G., GRENHAM, S., SCULLY, P., FITZGERALD, P., MOLONEY, R. D., SHANAHAN, F., DINAN, T. G. & CRYAN, J. F. 2013. The microbiome-gut-brain axis during early life regulates the hippocampal serotonergic system in a sex-dependent manner. Mol Psychiatry, 18, 666-73.

CORBILLÉ, A. G., PRETERRE, C., ROLLI-DERKINDEREN, M., CORON, E., NEUNLIST, M., LEBOUVIER, T. & DERKINDEREN, P. 2017. Biochemical analysis of α-synuclein extracted from control and Parkinson's disease colonic biopsies. Neurosci Lett, 641, 81-86.

DO VAN, B., GOUEL, F., JONNEAUX, A., TIMMERMAN, K., GELÉ, P., PÉTRAULT, M., BASTIDE, M., LALOUX, C., MOREAU, C., BORDET, R., DEVOS, D. & DEVEDJIAN, J. C. 2016. Ferroptosis, a newly characterized form of cell death in Parkinson's disease that is regulated by PKC. Neurobiol Dis, 94, 169-78.

EVANS, M. L., GICHANA, E., ZHOU, Y. & CHAPMAN, M. R. 2018. Bacterial Amyloids. Methods Mol Biol, 1779, 267-288.

FRIEDLAND, R. P. & CHAPMAN, M. R. 2017. The role of microbial amyloid in neurodegeneration. PLoS Pathog, 13,e1006654.

GALARIS, D., BARBOUTI, A. & PANTOPOULOS, K. 2019. Iron homeostasis and oxidative stress: An intimate relationship. Biochim Biophys Acta Mol Cell Res, 1866, 118535.

GUINEY, S. J., ADLARD, P. A., BUSH, A. I., FINKELSTEIN, D. I. & AYTON, S. 2017. Ferroptosis and cell death mechanisms in Parkinson's disease. Neurochem Int, 104, 34-48.

JIANG, H., WANG, J., ROGERS, J. & XIE, J. 2017. Brain Iron Metabolism Dysfunction in Parkinson's Disease. Mol Neurobiol, 54, 3078-3101.

KALIA, L. V. & LANG, A. E. 2015. Parkinson's disease. Lancet, 386, 896-912.

KATO, T., YAMAZAKI, K., NAKAJIMA, M., DATE, Y., KIKUCHI, J., HASE, K., OHNO, H. & YAMAZAKI, K. 2018. Oral Administration of Porphyromonas gingivalis Alters the Gut Microbiome and Serum Metabolome. mSphere, 3,e00460-18.

KELL, D. B. & PRETORIUS, E. 2014. Serum ferritin is an important inflammatory disease marker, as it is mainly a leakage product from damaged cells. Metallomics, 6, 748-73.

KELL, D. B. & PRETORIUS, E. 2018. No effects without causes: the Iron Dysregulation and Dormant Microbes hypothesis for chronic, inflammatory diseases. Biological Reviews, 93, 1518-1557.

KUMAR, D. K., CHOI, S. H., WASHICOSKY, K. J., EIMER, W. A., TUCKER, S., GHOFRANI, J., LEFKOWITZ, A., MCCOLL, G., GOLDSTEIN, L. E., TANZI, R. E. & MOIR, R. D. 2016. Amyloid-β peptide protects against microbial infection in mouse and worm models of Alzheimer's disease. Sci Transl Med, 8, 340ra72.

MADENCI, G., BILEN, S., ARLI, B., SAKA, M. & AK, F. 2012. Serum iron, vitamin B12 and folic acid levels in Parkinson's disease. Neurochem Res, 37, 1436-41.

MASALDAN, S., BUSH, A. I., DEVOS, D., ROLLAND, A. S. & MOREAU, C. 2019. Striking while the iron is hot: Iron metabolism and ferroptosis in neurodegeneration. Free Radic Biol Med, 133, 221-233.

MEDEIROS, M. S., SCHUMACHER-SCHUH, A., CARDOSO, A. M., BOCHI, G. V., BALDISSARELLI, J., KEGLER, A., SANTANA, D., CHAVES, C. M., SCHETINGER, M. R., MORESCO, R. N., RIEDER, C. R. & FIGHERA, M. R. 2016. Iron and Oxidative Stress in Parkinson's Disease: An Observational Study of Injury Biomarkers. PLoS One, 11, e0146129.

MOCHIZUKI, H., CHOONG, C. J. & BABA, K. 2020. Parkinson's disease and iron. J Neural Transm (Vienna), 127, 181-187.

NICOLETTI, G., CRESCIBENE, L., SCORNAIENCHI, M., BASTONE, L., BAGALÀ, A., NAPOLI, I. D., CARACCIOLO, M. & QUATTRONE, A. 2001. Plasma levels of vitamin E in Parkinson's disease. Arch Gerontol Geriatr, 33, 7-12.

PARK, S. C., MOON, J. C., SHIN, S. Y., SON, H., JUNG, Y. J., KIM, N. H., KIM, Y. M., JANG, M. K. & LEE, J. R. 2016. Functional characterization of alpha-synuclein protein with antimicrobial activity. Biochem Biophys Res Commun, 478, 924-8.

PLUM, S., STEINBACH, S., ATTEMS, J., KEERS, S., RIEDERER, P., GERLACH, M., MAY, C. & MARCUS, K. 2016. Proteomic characterization of neuromelanin granules isolated from human substantia nigra by laser-microdissection. Sci Rep, 6, 37139.

PRETORIUS, E., BESTER, J., VERMEULEN, N. & LIPINSKI, B. 2013a. Oxidation inhibits iron-induced blood coagulation. Curr Drug Targets, 14, 13-9.

PRETORIUS, E. & KELL, D. B. 2014. Diagnostic morphology: biophysical indicators for iron-driven inflammatory diseases. Integr Biol (Camb), 6, 486-510.

PRETORIUS, E., SWANEPOEL, A. C., BUYS, A. V., VERMEULEN, N., DUIM, W. & KELL, D. B. 2014. Eryptosis as a marker of Parkinson's disease. Aging (Albany NY), 6, 788.

PRETORIUS, E., VERMEULEN, N., BESTER, J., LIPINSKI, B. & KELL, D. B. 2013b. A novel method for assessing the role of iron and its functional chelation in fibrin fibril formation: the use of scanning electron microscopy. Toxicol Mech Methods, 23, 352-9.

QIU, Y., CAO, Y., CAO, W., JIA, Y. & LU, N. 2020. The Application of Ferroptosis in Diseases. Pharmacol Res, 159, 104919.

SOHMIYA, M., TANAKA, M., TAK, N. W., YANAGISAWA, M., TANINO, Y., SUZUKI, Y., OKAMOTO, K. & YAMAMOTO, Y. 2004. Redox status of plasma coenzyme Q10 indicates elevated systemic oxidative stress in Parkinson's disease. J Neurol Sci, 223, 161-6.

SUTACHAN, J. J., CASAS, Z., ALBARRACIN, S. L., STAB, B. R., 2ND, SAMUDIO, I., GONZALEZ, J., MORALES, L. & BARRETO, G. E. 2012. Cellular and molecular mechanisms of antioxidants in Parkinson's disease. Nutr Neurosci, 15, 120-6.

TOMLINSON, J. J., SHUTINOSKI, B., DONG, L., MENG, F., ELLEITHY, D., LENGACHER, N. A., NGUYEN, A. P., CRON, G. O., JIANG, Q., ROBERSON, E. D., NUSSBAUM, R. L., MAJBOUR, N. K., EL-AGNAF, O. M., BENNETT, S. A., LAGACE, D. C., WOULFE, J. M., SAD, S., BROWN, E. G. & SCHLOSSMACHER, M. G. 2017. Holocranohistochemistry enables the visualization of α-synuclein expression in the murine olfactory system and discovery of its systemic anti-microbial effects. J Neural Transm (Vienna), 124, 721-738.

WEILAND, A., WANG, Y., WU, W., LAN, X., HAN, X., LI, Q. & WANG, J. 2019. Ferroptosis and Its Role in Diverse Brain Diseases. Mol Neurobiol, 56, 4880-4893.

YU, H., GUO, P., XIE, X., WANG, Y. & CHEN, G. 2017. Ferroptosis, a new form of cell death, and its relationships with tumourous diseases. J Cell Mol Med, 21, 648-657.

Round 2

Reviewer 1 Report

- The new title better reflects the content of the manuscript, although the term “iron dysregulation” is now missing from the title and seems important later on. Maybe could be added “Iron dysregulation and inflammagens related to …. Are central to the development of PD”.

- Section 2: These sentences seem repetitive “Both pathological levels of iron and oxidative stress are of importance in both neuro- as well as systemic inflammation in PD patients. The spectrum of iron dysregulation in PD includes iron storage, uptake and release [53]; and importantly, the effects may  be seen both in the brain and also the peripheral tissues. Iron dysregulation in PD, occurs both in the brain and in circulation (Figure 1) (for a review see [54]).” Please revise.

- I would suggest to move the following sentence “Neuromelanin is a complex polymer pigment found primarily 182 in the dopaminergic neurons of human substantia nigra, and it is stored in granules including a 183 protein matrix and lipid droplets [73]” after this one: “An important iron compound in dopamine and 178 norepinephrine neurons is the neuromelanin-iron complex [53, 71].”

- The title from section 4.3. was correct, I meant to change the title of section 4 to include Oral and gut dysbiosis.

Author Response

REVIEWER 1 COMMENTS:

The new title better reflects the content of the manuscript, although the term “iron dysregulation” is now missing from the title and seems important later on. Maybe could be added “Iron dysregulation and inflammagens related to …. Are central to the development of PD”.

Thank you we now changed the title to read:

Iron dysregulation and inflammagens related to oral and gut health are central to the development of Parkinson’s disease

- Section 2: These sentences seem repetitive “Both pathological levels of iron and oxidative stress are of importance in both neuro- as well as systemic inflammation in PD patients. The spectrum of iron dysregulation in PD includes iron storage, uptake and release [53]; and importantly, the effects may  be seen both in the brain and also the peripheral tissues. Iron dysregulation in PD, occurs both in the brain and in circulation (Figure 1) (for a review see [54]).” Please revise.

We have removed the sentence:

Both pathological levels of iron and oxidative stress are of importance in both neuro- as well as systemic inflammation in PD patients.

- I would suggest to move the following sentence “Neuromelanin is a complex polymer pigment found primarily 182 in the dopaminergic neurons of human substantia nigra, and it is stored in granules including a 183 protein matrix and lipid droplets [73]” after this one: “An important iron compound in dopamine and 178 norepinephrine neurons is the neuromelanin-iron complex [53, 71].”

OK thank you, it does read much better!

- The title from section 4.3. was correct, I meant to change the title of section 4 to include Oral and gut dysbiosis.

Apologies, it is now changed and 4.3 restored to the original title.